# Revisiting Sparse Convolutional Model for Visual Recognition

**Xili Dai**[1]* **Mingyang Li**[2]* **Pengyuan Zhai**[3] **Shengbang Tong**[4] **Xingjian Gao**[4]
**Shao-Lun Huang**[2] **Zhihui Zhu**[5] **Chong You**[4] **Yi Ma**[2,4]

[1]The Hong Kong University of Science and Technology (Guangzhou)
[2]Tsinghua-Berkeley Shenzhen Institute (TBSI), Tsinghua University
[3] Harvard University    [4] University of California, Berkeley    [5] Ohio State University

## Abstract

Despite strong empirical performance for image classification, deep neural networks are often regarded as "black boxes" and they are difficult to interpret. On the other hand, sparse convolutional models, which assume that a signal can be expressed by a linear combination of a few elements from a convolutional dictionary, are powerful tools for analyzing natural images with good theoretical interpretability and biological plausibility. However, such principled models have not demonstrated competitive performance when compared with empirically designed deep networks. This paper revisits the sparse convolutional modeling for image classification and bridges the gap between good empirical performance (of deep learning) and good interpretability (of sparse convolutional models). Our method uses differentiable optimization layers that are defined from convolutional sparse coding as drop-in replacements of standard convolutional layers in conventional deep neural networks. We show that such models have equally strong empirical performance on CIFAR-10, CIFAR-100 and ImageNet datasets when compared to conventional neural networks. By leveraging stable recovery property of sparse modeling, we further show that such models can be much more robust to input corruptions as well as adversarial perturbations in testing through a simple proper trade-off between sparse regularization and data reconstruction terms. Source code can be found at `https://github.com/Delay-Xili/SDNet`.

## 1 Introduction

In recent years, deep learning has been a dominant approach for image classification and has significantly advanced the performance over previous shallow models. Despite the phenomenal empirical success, it has been increasingly realized as well as criticized that deep convolutional networks (ConvNets) are "black boxes" for which we are yet to develop clear understanding [1]. The layer operations such as convolution, nonlinearity and normalization are geared towards minimizing an end-to-end training loss and *do not have much data-specific meaning*. As such, the functionality of each intermediate layer in a trained ConvNets is mostly unclear and the feature maps that they produce are hard to interpret. The lack of interpretability also contributes to the notorious difficulty in enhancing such learning systems for practical data which are usually corrupted by various forms of perturbation.

This paper presents a visual recognition framework by introducing layers that *have explicit data modeling* to tackle shortcomings of current deep learning systems. We work under the assumption that the layer input can be represented by *a few* atoms from a dictionary shared by all data points. This

---

*Equal contribution

36th Conference on Neural Information Processing Systems (NeurIPS 2022).

is the classical *sparse* data modeling that, as shown in a pioneering work of [2], can easily discover meaningful structures from natural image patches. Backed by its ability in learning interpretable representations and strong theoretical guarantees [3, 4, 5, 6, 7, 8] (e.g. for handling corrupted data), sparse modeling has been used broadly in many signal and image processing applications [9]. However, the empirical performance of sparse methods have been surpassed by deep learning methods for classification of modern image datasets.

Because of the complementary benefits of sparse modeling and deep learning, there exist many efforts that leverage sparse modeling to gain theoretical insights into ConvNets and/or to develop computational methods that further improve upon existing ConvNets. One of the pioneering works is [10] which interpreted a ConvNet as approximately solving a multi-layer convolutional sparse coding model. Based on this interpretation, the work [10] and its follow-ups [11, 12, 13, 14] presented alternative algorithms and models in order to further enhance the practical performance of such learning systems. However, there has been no empirical evidence that such systems can handle modern image datasets such as ImageNet and obtain comparable performance to deep learning. The only exception to the best of our knowledge is the work of [15, 16] which exhibited a performance on par to (on ImageNet) or better than (on CIFAR-10) ResNet. However, the method in [15, 16] 1) requires a dedicated design of network architecture that may limit its applicability, 2) is computationally orders of magnitude slower to train, and 3) does not demonstrate benefits in terms of interpretability and robustness. In a nutshell, sparse modeling is yet to demonstrate practicality that enables its broad applications.

**Paper contributions.** In this paper, we revisit sparse modeling for image classification and demonstrate through a simple design that sparse modeling can be combined with deep learning to obtain performance on par with standard ConvNets but with better layer-wise interpretability and stability. Our method encapsulates the sparse modeling into an *implicit layer* [17, 18, 19] and uses it as a drop-in replacement for any convolutional layer in standard ConvNets. The layer implements the convolutional sparse coding (CSC) model of [20], and is referred to as a *CSC-layer*, where the input signal is approximated by a sparse linear combination of atoms from a convolutional dictionary. Such a convolutional dictionary is treated as the parameters of the CSC-layer that are amenable to training via back-propagation. Then, the overall network with the CSC-layers may be trained in an end-to-end fashion from labeled data by minimizing the cross-entropy loss as usual. This paper demonstrates that such a learning framework has the following benefits:

- **Performance on standard datasets.** We demonstrate that our network obtains better (on CIFAR-100) or on par (on CIFAR-10 and ImageNet) performance with similar training time compared with standard architectures such as ResNet [21]. This provides the first evidence on the strong empirical performance of sparse modeling for deep learning to the best of our knowledge. Compared to previous sparse methods [15] that obtained similar performance, our method is of orders of magnitude faster.

- **Robustness to input perturbations.** The stable recovery property of sparse convolution model equips the CSC-layers with the ability to remove perturbation in the layer input and to recover clean sparse code. As a result, our networks with CSC-layers are more robust to perturbations in the input images compared with classical neural networks. Unlike existing approaches for obtaining robustness that require heavy data augmentation [22] or additional training techniques [23], our method is light-weight and does not require modifying the training procedure at all.

## 2 Related Work

**Implicit layers.** The idea of trainable layers defined from implicit functions can be traced back at least to the work of [24]. Recently, there is a revival of interests in implicit layers [17, 18, 19, 25, 26, 27, 28, 29] as an attractive alternative to explicit layers in existing neural networks. However, a majority of the cited works above define an implicit layer by a fixed point iteration, typically motivated from existing explicit layers such as residual layers, therefore they do not have clear interpretation in terms of modeling of the layer input. Consequently, such models do not have the ability to deal with input perturbations. The only exceptions are differentiable optimization layers [30, 17, 18, 31] that incorporate complex dependencies between hidden layers through the formulation of convex optimization. Nevertheless, most of the above works focus on differentiating through the convex optimization layers (such as disciplined parametrized programming [18]) without specializing in any

particular signal models such as the sparse models considered in this paper nor demonstrating their performance when encapsulated in multi-layer neural networks.

**Sparse prior in deep learning.** Aside from image classification, sparse modeling has been introduced to deep learning for many image processing tasks such as super-resolution [32], denoising [33] and so on [34, 35, 36, 37]. These works incorporate sparse modeling by using network architectures that are *motivated by* (but are not the same as) an unrolled sparse coding algorithm LISTA [38]. In sharp contrast to ours, there is no guarantee that such architectures perform a sparse encoding with respect to a particular (convolution) dictionary at all. As a result, they lack the capability of handling input perturbations as in our method. A notable exception is the work of [15] where each layer performs a precise sparse encoding and exhibits on par or better performance for image classification over ResNet. However, the practical benefit of the sparse modeling in terms of robustness is not demonstrated. Moreover, [15] adopts a patch-based sparse coding model for images and has a large computational burden.

**Robustness.** It is known that modern neural networks are extremely vulnerable to small perturbations in the input data. A plethora of techniques have been proposed to address this instability issue, including stability training [23], adversarial training [39, 40, 41], data augmentation [42, 43, 22], etc. Nevertheless, these techniques either need a computational and memory overhead, or require a selection of appropriate augmentation strategies to cover all possible corruptions. With standard training only, our model can be made robust to input perturbations in test data by simply adapting sparse modeling to account for noise. Closely related to our work are [44, 45, 46] which use sparse modeling to improve adversarial robustness. However, they either only demonstrate performance on very simple networks [45, 46] or sacrifice natural accuracy for robustness [44]. In contrast, our method is tested on realistic networks and does not affect natural accuracy.

## 3 Neural Networks with Sparse Modeling

In this section, we show how sparse modeling is incorporated into a deep network via a specific type of network layer that we refer to as the convolutional sparse coding (CSC) layer. We describe the CSC-layer in Sec. 3.1 and explain how we use them for deep learning in Sec. 3.2. Finally, Sec. 3.3 explains how CSC enables robust inference with corrupted test data.

**Notations.** Given a single-channel image $\boldsymbol{\xi} \in \mathbb{R}^{H \times W}$ represented as a matrix, we may treat it as a 2D signal defined on the discrete domain $[1, \ldots, H] \times [1, \ldots, W]$ and extended to $\mathbb{Z} \times \mathbb{Z}$ by padding zeros. Given a 2D kernel $\boldsymbol{\alpha} \in \mathbb{R}^{k \times k}$, we may treat it as a 2D signal defined on the discrete domain $[-k_0 \cdots, k_0] \times [-k_0, \cdots, k_0]$ with $k = 2k_0 + 1$ and extended to $\mathbb{Z} \times \mathbb{Z}$ by padding zeros. Then, for convenience, we use "$*$" and "$\star$" to denote the convolution and correlation operators, respectively, between two 2D signals:

$$
\begin{aligned}
(\boldsymbol{\alpha} * \boldsymbol{\xi})[i,j] &\doteq \sum_p \sum_q \boldsymbol{\xi}[i-p, j-q] \cdot \boldsymbol{\alpha}[p,q], \\
(\boldsymbol{\alpha} \star \boldsymbol{\xi})[i,j] &\doteq \sum_p \sum_q \boldsymbol{\xi}[i+p, j+q] \cdot \boldsymbol{\alpha}[p,q].
\end{aligned}
\tag{1}
$$

### 3.1 Convolutional Sparse Coding (CSC) Layer

Sparse modeling is introduced in the form of an *implicit layer* of a neural network. Unlike classical fully-connected or convolutional layers in which input-output relations are defined by an explicit function, implicit layers are defined from implicit functions. For our case, in particular, we introduce an implicit layer that is defined from an optimization problem involving the input to the layer as well as a weight parameter, where the output of the layer is the solution to the optimization problem.

**A generative model via sparse convolution.** Concretely, given a multi-dimensional input signal $\boldsymbol{x} \in \mathbb{R}^{M \times H \times W}$ to the layer where $H, W$ are spatial dimensions and $M$ is the number of channels for $\boldsymbol{x}$. We assume the signal $\boldsymbol{x}$ is generated by a multi-channel sparse code $\boldsymbol{z} \in \mathbb{R}^{C \times H \times W}$ convoluting with a multi-dimensional kernel $\boldsymbol{A} \in \mathbb{R}^{M \times C \times k \times k}$, which is referred to as a convolution *dictionary*. Here $C$ is the number of channels for $\boldsymbol{z}$ and the convolution kernel $\boldsymbol{A}$. To be more precise, we denote

$z$ as $z \doteq (\zeta_1, \dots, \zeta_C)$ where each $\zeta_c \in \mathbb{R}^{H \times W}$ (presumably sparse), and denote the kernel $A$ as

$$A \doteq \begin{pmatrix} \alpha_{11} & \alpha_{12} & \alpha_{13} & \cdots & \alpha_{1C} \\ \alpha_{21} & \alpha_{22} & \alpha_{23} & \cdots & \alpha_{2C} \\ \vdots & \vdots & \vdots & \ddots & \vdots \\ \alpha_{M1} & \alpha_{M2} & \alpha_{M3} & \cdots & \alpha_{MC} \end{pmatrix} \quad \in \mathbb{R}^{M \times C \times k \times k}, \tag{2}$$

where each $\alpha_{ij} \in \mathbb{R}^{k \times k}$ is a kernel of size $k \times k$. Then the signal $x$ is generated via the following operator $\mathcal{A}(\cdot)$ defined by the kernel $A$ as:

$$x = \mathcal{A}(z) \doteq \sum_{c=1}^{C} \left( \alpha_{1c} \star \zeta_c, \dots, \alpha_{Mc} \star \zeta_c \right) \quad \in \mathbb{R}^{M \times H \times W}. \tag{3}$$

**A layer as convolutional sparse coding.** Given a multi-dimensional input signal $x \in \mathbb{R}^{M \times H \times W}$, we define that the function of "a layer" is to perform an (inverse) mapping to a preferably sparse output $z_* \in \mathbb{R}^{C \times H \times W}$, where $C$ is the number of output channels. Under the above sparse generative model, we can seek the optimal sparse solution $z$ by solving the following Lasso type optimization problem:

$$z_* = \arg\min_z \lambda \|z\|_1 + \frac{1}{2}\|x - \mathcal{A}(z)\|_2^2 \quad \in \mathbb{R}^{C \times H \times W}. \tag{4}$$

The optimization problem in (4) is based on the convolutional sparse coding (CSC) model [20][2]. Hence, we refer to the implicit layer defined by (4) as a *CSC-layer*. The goal of the CSC model is to reconstruct the input $x$ via $\mathcal{A}(z)$, where the feature map $z$ specifies the locations and magnitudes of the convolutional filters in $A$ to be linearly combined (see Figure 1 for an illustration). The reconstruction is not required to be exact in order to tolerate modeling discrepancies, and the difference between $x$ and $\mathcal{A}(z)$ is penalized by its entry-wise $\ell_2$-norm (i.e., the $\ell_2$ norm of $x - \mathcal{A}(z)$ flattened into a vector). Sparse modeling is introduced by the entry-wise $\ell_1$-norm of $z$ in the objective function, which enforces $z$

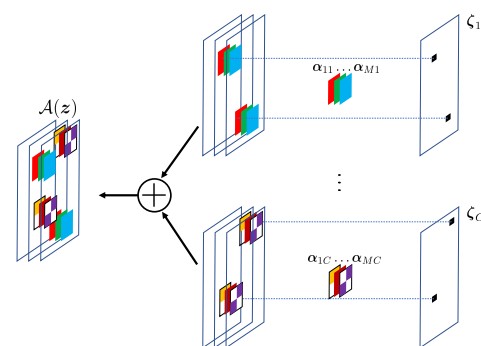

Figure 1: Illustration of the operator $\mathcal{A}$ in the convolutional sparse coding model for the CSC-layer.

to be sparse. The parameter $\lambda > 0$ controls the tradeoff between the sparsity of $z$ and the magnitude of the residual $x - \mathcal{A}(z)$, and is treated as a hyper-parameter that subjects to tuning via cross-validation. As we will show in Sec. 3.3, $\lambda$ can be used to improve the performance of our model in the test phase when the input is corrupted.

Based on the input-output mapping of the CSC-layer given in (4), one may perform forward propagation by solving the associated optimization, and perform backward propagation by deriving the gradient of $z_*$ with respect to the input $x$ and parameter $A$. In this paper, we adopt the fast iterative shrinkage thresholding algorithm (FISTA) [47] for the forward propagation, which also produces an unrolled network architecture that can carry out automatic differentiation for backward propagation. We defer a discussion of the implementation details of the CSC-layer to the Appendix.

### 3.2 Sparse Dictionary Learning Network Architecture and Training

Convolution layers are basic ingredients of ConvNets that appear in many common network architectures such as LeNet [48] and ResNet [49]. In this paper, we incorporate sparse modeling into a given existing/baseline network architecture by replacing certain / all convolution layers with the CSC-layer. Meanwhile, all other layers such as normalization, nonlinear, and fully connected layers

---

[2]Typically, convolution operators "$*$" are used in the definition of the operator $\mathcal{A}$ (see (3)), rather than the correlation operators "$\star$". We adopt the definition in (3) to be consistent with the convention of modern deep learning packages.

are retained. This simple design choice allows the CSC-layers to be broadly applicable. We will refer to our network with CSC-layers as *Sparse Dictionary learning Network* (SDNet).

Give a set of training data $\{\boldsymbol{x}_i, \boldsymbol{y}_i\}_{i=1}^N$ where $\boldsymbol{x}_i$ denotes an image and $\boldsymbol{y}_i$ is the corresponding label, we train a network with CSC-layers by solving the following optimization problem:

$$\min_{\boldsymbol{\theta}} \frac{1}{N} \sum_{i=1}^N \ell_{\text{CE}}\Big(f(\boldsymbol{x}_i; \boldsymbol{\theta}), \boldsymbol{y}_i\Big) \quad \text{s.t. } \boldsymbol{A}_s \in \mathcal{N} \ \forall s \in \mathcal{S}, \tag{5}$$

where $\ell_{\text{CE}}$ denotes the cross-entropy loss. In above, we use $f(\cdot; \boldsymbol{\theta})$ to denote the mapping that is performed by the neural network, where $\boldsymbol{\theta}$ is a set of learnable parameters containing a subset of kernel parameters $\{\boldsymbol{A}_s\}_{s \in \mathcal{S}}$ associated with the set $\mathcal{S}$ of CSC-layers. Following the convention in the sparse dictionary learning literature, we add the constraint $\boldsymbol{A}_s \in \mathcal{N}$ for the dictionaries in CSC-layers, where $\mathcal{N}$ denotes the set of normalized dictionaries:

$$\mathcal{N} \doteq \left\{ \boldsymbol{A} \in \mathbb{R}^{M \times C \times k \times k} : \sum_{m=1}^M \|\boldsymbol{\alpha}_{mc}\|_2^2 = 1, \forall c \in [C] \right\}.$$

To handle such a constraint, we use the projected stochastic gradient descent (SGD) for solving the problem in (5). That is, after each gradient update step for the parameters $\boldsymbol{\theta}$ and $\{\boldsymbol{A}_s\}_{s \in \mathcal{S}}$ as in a regular SGD, an extra step is taken to project each $\boldsymbol{A}_s$ onto the constraint set $\mathcal{N}$.

### 3.3 Robust Inference

The fundamental difference of the CSC-layer vis-a-vis a classical explicit layer (e.g., a convolutional layer) is that the CSC-layer imposes an assumption on the input feature map. That is, it assumes that the input feature map (or image) can be approximated by a superposition of a few atoms of a dictionary, which is the layer parameter that is learnable and is shared across all data. In this section, we show that CSC-layers enables us to design a robust inference strategy to obtain robustness to corruptions in ways that cannot be achieved by classical explicit layers.

We leverage an attractive property of the CSC model is that it admits a stable recovery of the sparse signals with respect to input noise.

In particular, Theorem 1 shows that *any* bounded perturbation to the input of a CSC-layer produces a bounded perturbation to its output and does not change the support of the output, if one uses a $\lambda$ in (4) that is proportional to the norm of the perturbation in the layer input, provided that certain technical conditions are satisfied.

**Theorem 1.** *(Informal version of [50, Theorem 19]) Suppose $\boldsymbol{x}_\natural$ has a representation $\mathcal{A}(\boldsymbol{z}_\natural)$ as in (3), and that it is contaminated by noise $\boldsymbol{e}$ to create the input $\boldsymbol{x} = \boldsymbol{x}_\natural + \boldsymbol{e}$. Then as long as $\boldsymbol{z}_\natural$ is sufficiently sparse, the solution $\boldsymbol{z}_*$ to (4) with $\lambda = O(\|\boldsymbol{e}\|_2)$ satisfies $(i)$ the support of $\boldsymbol{z}_*$ is contained in that of $\boldsymbol{z}_\natural$ and $(ii) \|\boldsymbol{z}_* - \boldsymbol{z}_\natural\|_2 = O(\|\boldsymbol{e}\|_2)$.*

Intuitively, the parameter $\lambda$ in (4) controls a balance between the sparsity regularization $\|\boldsymbol{z}\|_1$ and the residual $\boldsymbol{x} - \mathcal{A}(\boldsymbol{z})$, the latter of which accounts for modeling discrepancies and increases when $\boldsymbol{x}$ is noisy. Therefore, using a larger value of $\lambda$ helps the model to handle a larger residual.

We thus present a very simple approach to obtain model robustness. We consider the setting where a model is trained on an uncorrupted dataset but is tested on data that is corrupted by random noise. Corruptions in the input image may propagate into deeper feature maps (hence corrupting the input of all CSC-layers) during forward propagation. Therefore, instead of directly using the CSC-layers that are obtained from the training phase, our method is to adjust the trade-off parameter $\lambda$ in the CSC-layers for the test data. As we show in the experiments (see Sec. 4.2), an optimal value of $\lambda$ indeed increases with the variance of the noise in the input image, which is well-aligned with the result in Theorem 1.

**Choosing the optimal $\lambda$.** In practical applications, the amount of noise in a given test dataset is often unknown. Hence, the optimal choice of parameter $\lambda$ becomes a nontrivial task. Here we present a practical technique for determining a proper choice of value $\lambda$, based on the simple observation that the amount of noise in the test data correlates with the magnitude of the residual $\boldsymbol{x} - \mathcal{A}(\boldsymbol{z})$. That is, for test data corrupted by a larger amount of noise, we expect the magnitude of the residual in CSC-layers to become larger with such data fed into the network. Since the residual for any test data

**Algorithm 1** Robust inference with neural networks constructed from CSC-layers

---

**Input:** A network architecture with CSC-layers $f(\cdot; \boldsymbol{\theta}, \lambda_0)$, a (clean) training data $\mathcal{T}_{\text{train}}$, a (corrupted) test data $\mathcal{T}_{\text{test}}$, corruption type T, a set $\mathcal{C}$ of corruption levels, a set $\Lambda$ of values for $\lambda$.

1: *# Training the network*
2: Train the network $f(\cdot; \boldsymbol{\theta}, \lambda_0)$ on $\mathcal{T}_{\text{train}}$ as described in Sec. 3.2 to obtain learned parameters $\boldsymbol{\theta}_{\star}$.
3: *# Fitting a relationship between optimal $\lambda$ and the residual from CSC-layers using $\mathcal{T}_{\text{train}}$*
4: **for** each noise level $c \in \mathcal{C}$ **do**
5:     Generate corrupted data $\mathcal{T}_{\text{train}}^c$ by injecting random noise of type T with level $c$ to $\mathcal{T}_{\text{train}}$.
6:     Apply $f(\cdot; \boldsymbol{\theta}_{\star}, \lambda_0)$ on $\mathcal{T}_{\text{train}}^c$ and compute averaged residual from all CSC-layers as $r_c$.
7:     **for** each parameter $\lambda \in \Lambda$ **do**
8:         Apply $f(\cdot; \boldsymbol{\theta}_{\star}, \lambda)$ on $\mathcal{T}_{\text{train}}^c$ and compute averaged accuracy as $a_\lambda$.
9:     **end for**
10:     Set $\lambda_c = \arg\max_{\lambda \in \Lambda} a_\lambda$.
11: **end for**
12: Fit a function $\lambda := \lambda(r)$ from $\{\lambda_c, r_c\}_{c \in \mathcal{C}}$ via linear least squares.
13: *# Computing the residual from CSC-layers on $\mathcal{T}_{\text{test}}$*
14: Apply $f(\cdot; \boldsymbol{\theta}_{\star}, \lambda_0)$ on $\mathcal{T}_{\text{test}}$ and compute averaged residual from all CSC-layers as $r_{\text{test}}$.

**Output:** Predicted labels on $\mathcal{T}_{\text{test}}$ with the network $f(\cdot; \boldsymbol{\theta}_{\star}, \lambda(r_{\text{test}}))$.

---

can always be computed, the key question here is how we can find a relationship between an optimal value $\lambda$ and the magnitude of the residual.

Our technique for addressing this challenge is to learn such a relationship on the training set, by injecting synthetic data corruptions. For simplicity we summarize our technique in Algorithm 1, and explain it in details below. We assume that although the amount of noise in a test dataset $\mathcal{T}_{\text{test}}$ is unknown, the type of the noise (e.g., Gaussian noise, shot noise, etc.) is known. Given the noise type, we first determine a set $\mathcal{C}$ which contains a set of values specifying the potential amount of noise in test data. For example, if we consider Gaussian noise, then $\mathcal{C}$ contains a set of values specifying the variance of the noise. We also specify a set $\Lambda$ of potential values for $\lambda$ to be used for inference. After training the network as specified in Step 2, we use a procedure described in between Step 4 and Step 12 to fit a function that maps a residual value $r$ to an optimal choice of $\lambda$. The idea for fitting such a relation is to generate synthetic noise of varying magnitudes in $\mathcal{C}$ on the training set, and for each noise magnitude we sweep the parameter $\lambda \in \Lambda$ to find a $\lambda$ that produces the best accuracy on training set. Once such a relationship is learned, we feed in the residual value computed on the test dataset $\mathcal{T}_{\text{test}}$ to predict an optimal $\lambda$ to be used for inference on test data, as described in Step 14.

## 4 Experiments

In this section, we provide experimental evidence for neural networks with CSC-layers as discussed in Sec. 3. Through experiments on CIFAR-10, CIFAR-100[3], and ImageNet[4], Sec. 4.1 shows that our networks have equally competitive classification performance as mainstream architectures such as ResNet. Furthermore, we show in Sec. 4.2 that our network is able to handle input perturbations with the robust inference technique. Finally, we demonstrate in Sec. 4.3 that our network is able to handle adversarial perturbations as well. More details about implementation are given in the Appendix.

**Datasets.** We test the performance of our method using the CIFAR-10 and CIFAR-100 [51] datasets. Each dataset contains 50,000 training images and 10,000 testing images where each image is of size $32 \times 32$ with RGB channels. We also use the ImageNet dataset [52] that contains 1,000 classes and a total number of around 1 million images.

**Network architecture.** We use the network architectures with the first convolutional layers of ResNet-18 and ResNet-34 [21][5] replaced by CSC-layers, and refer to these networks as SDNet-18 and SDNet-34, respectively. We use $\lambda = 0.1$ as the trade-off parameter in (4) for all CSC-layers

---

[3]CIFAR-10 and CIFAR-100 are available at `https://www.cs.toronto.edu/~kriz/cifar.html`
[4]ImageNet is a publicly available dataset: `https://www.image-net.org`
[5]We use the implementation at `https://github.com/kuangliu/pytorch-cifar`, which is under the MIT License with Copyright (c) 2017 liukuang.

Table 1: Comparison of different network archtectures, including ResNet, Multi-scale Deep Equilibrium (MDEQ), Sparse Coding Network (SCN, SCN-first), and our SDNet, for image classification tasks. We report the number of model parameters (i.e., Model Size), accuracy on test data (i.e., Top-1 Acc), GPU memory consumption during training (i.e., Memory), and the number of images that are handled per second (n/s) during training (i.e., Speed).

| Dataset | Architecture | Model Size | Top-1 Acc | Memory | Speed |
|---|---|---|---|---|---|
| CIFAR-10 | ResNet-18 [21] | 11.2M | 95.54% | 1.0 GB | 1600 n/s |
| | ResNet-34 [21] | 21.1M | 95.57% | 2.0 GB | 1000 n/s |
| | MDEQ [27] | 11.1M | 93.80% | 2.0 GB | 90 n/s |
| | SCN [15] | 0.7M | 94.36% | 10.0GB | 39 n/s |
| | SCN-18 | 11.2M | 95.12% | 3.5 GB | 158 n/s |
| | SDNet-18 (ours) | 11.2M | 95.20% | 1.2 GB | 1500 n/s |
| | SDNet-34 (ours) | 21.1M | 95.57% | 2.4 GB | 900 n/s |
| CIFAR-100 | ResNet-18 [21] | 11.2M | 77.82% | 1.0 GB | 1600 n/s |
| | ResNet-34 [21] | 21.1M | 78.39% | 2.0 GB | 1000 n/s |
| | MDEQ [27] | 11.2M | 74.12% | 2.0 GB | 90 n/s |
| | SCN [15] | 0.7M | 80.07% | 10.0GB | 39 n/s |
| | SCN-18 | 11.2M | 78.59% | 3.5 GB | 158 n/s |
| | SDNet-18 (ours) | 11.3M | 78.31% | 1.2 GB | 1500 n/s |
| | SDNet-34 (ours) | 21.2M | 78.48% | 2.4 GB | 900 n/s |
| ImageNet | ResNet-18 [21] | 11.7M | 68.98% | 24.1 GB | 2100 n/s |
| | ResNet-34 [21] | 21.5M | 72.83% | 32.3 GB | 1400 n/s |
| | SCN [15] | 9.8M | 70.42% | 95.1 GB | 51 n/s |
| | SDNet-18 (ours) | 11.7M | 69.47% | 37.6 GB | 1800 n/s |
| | SDNet-34 (ours) | 21.5M | 72.67% | 46.4 GB | 1200 n/s |

unless specified otherwise. Forward propagation through each CSC-layer is performed via unrolling two iterations of FISTA.

**Network training.** For CIFAR-10 and CIFAR-100, we use a cosine learning rate decay schedule with an initial learning rate of 0.1, and train the model for 220 epochs. We use the SGD optimizer with 0.9 momentum and Nestrov. The weight decay is set to $5 \times 10^{-4}$, and batch size is set to 128. All the experiments are conducted on a single NVIDIA GTX 2080Ti GPU. For ImageNet, we use multi-step learning rate decay schedule with an initial learning rate of 0.1 decayed by a factor of 0.1 at the 30th, 60th, and 90th epochs, and train the model for 100 epochs. The batch size is 512, and the optimizer is SGD with 0.9 momentum and Nestrov. All experiments on ImageNet are conducted on 4 NVIDIA RTX 3090 GPUs.

## 4.1 Performance for Image Classification

We compare our method with standard network architectures ResNet-18 and ResNet-34 [21]. In addition, we compare architectures with implicit layers (i.e., MDEQ [27]) and architectures with sparse modeling (i.e., SCN [15]). For ResNet-18 and ResNet-34, we train the model using the same setup as our SDNet models. For MDEG and SCN, we train the models using the settings as stated in their respective papers. SCN has both a different sparse coding layer and a different network architecture compared to our SDNet. Hence, we also include a baseline referred to as SCN-18, which is constructed by replacing the first convolutional layer of ResNet-18 with the sparse coding layer from SCN (hence has the same architecture as SDNet-18 but a different sparse coding layer).

The results are reported in Table 1. We see that with similar model size, SDNet-18/34 produces a Top-1 accuracy that closely matches (for CIFAR-10 and ImageNet) or surpasses (for CIFAR-100) that of ResNet-18/34 while having a comparable speed. This shows the potential of our network with modeling based layers as a powerful alternative to existing data-driven models, since our model has the additional benefit of handling corruptions.

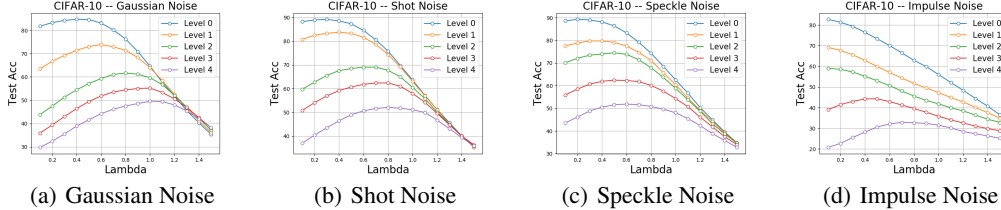

|  |  |  |  |
|:---:|:---:|:---:|:---:|
| (a) Gaussian Noise | (b) Shot Noise | (c) Speckle Noise | (d) Impulse Noise |

Figure 2: Test accuracy of SDNet-18 trained on CIFAR-10 dataset with $\lambda = 0.1$ and evaluated on 4 types of additive noise from CIFAR-10-C [53] in 5 severity levels each with varying values of $\lambda$. For each corruption type, optimal value of $\lambda$ for testing increases monotonically with the severity level.

We also compare our SDNet-18 model with the MDEQ model, which has a similar model size, and see that SDNet-18 is not only more accurate than MDEQ but is much ($> 7$ times) faster. Note that MDEQ cannot handle corrupted data as in our method as well.

The SCN network, which also uses sparse modeling, obtains a Top-1 accuracy that is highly competitive to all methods. However, a significant drawback of SCN is that it is very slow to train. This is true even with SCN-18, where only one convolutional layer is replaced by the sparse coding layer. The reason may be that SCN uses a patch-based sparse coding model for images, in contrast to a convolutional sparse coding model as in our method, which requires solving many sparse coding problems in each forward propagation that cannot benefit from parallel computing.

## 4.2 Handling Input Perturbations

To test the robustness of our method to input perturbations, we use the CIFAR-10-C dataset [53] which contains a test set for CIFAR-10 that is corrupted with different types of synthetic noise and 5 severity levels for each type. Because the CSC model in (4) penalizes the entry-wise difference between input and reconstructed signals, it is more suited for handling additive noises. Hence, we focus on four types of additive noises in CIFAR-10-C, namely, Gaussian noise, shot noise, speckle noise, and impulse noise. We evaluate the accuracy of our SDNet-18 and compare its performance with ResNet-18.

**Robustness as a function of $\lambda$.** As discussed in Sec. 3.3, we may improve the performance of our model to noisy test data by using values of $\lambda$ that are different from the training phase. Therefore, we report the performance of our method with varying $\lambda$ in the range of $[0.1, 1.5]$ in Figure 2 (recall that $\lambda = 0.1$ is used for training). It can be seen that for all types of noises and all severity levels (except for impulse noise with levels 0, 1, and 2), properly choosing a value of $\lambda$ that is different from that used during training helps to improve the test performance. In particular, the accuracy curves as a function of $\lambda$ exhibit a unimodal shape where the performance first increases and then decreases. Moreover, within each corruption type the values of $\lambda$ where a peak performance is achieved increase monotonically with the severity level of the corruption. Such an observation is well-aligned with our discussion in Sec. 3.3.

**Choosing an optimal $\lambda$.** While Figure 2 demonstrates that one may improve the performance on corrupted data via a proper choice of $\lambda$, it does not show how to choose the best $\lambda$ in practice. Here we show that the technique presented in Algorithm 1 can be used to select $\lambda$ for robust inference. Specifically, we apply Algorithm 1 with $f(\cdot; \boldsymbol{\theta}, \lambda_0)$ being SDNet-18 with $\lambda_0 = 0.1$, $\mathcal{T}_{\text{train}}$ being the CIFAR-10 training set, $\mathcal{T}_{\text{test}}$ being CIFAR-10-C data with a particular type of corruption under a particular severity level, and report the performance of the algorithm output in Table 2 and Table 3.

Table 2 shows the results with varying severity levels of Gaussian noise. We can see that if we use the same $\lambda = 0.1$ as in training (i.e., SDNet-18 w/ $\lambda = 0.1$), then the performance of SDNet-18 is already significantly better ResNet-18. This may be attributed to the fact that the CSC-layer is intrinsically more robust to input perturbations than convolution layers owning to its sparse modeling. However, if we use an adaptive $\lambda$ computed from Algorithm 1 (i.e., SDNet-18 w/ adaptive $\lambda$), then the performance is further significantly improved at all noise levels. To demonstrate that our method works beyond Gaussian noise, we report results for each of the four types of additive noises averaged over all severity levels in Table 3 for CIFAR-10-C as well as ImageNet-C. The results demonstrate

Table 2: Classification result under varying severity levels of Gaussian noise on CIFAR-10-C with $\lambda = 0.1$ (i.e., same as training) and with an adaptive $\lambda$ computed from Algorithm 1.

| Severity Level | Level-0 | Level-1 | Level-2 | Level-3 | Level-4 |
|---|---|---|---|---|---|
| ResNet-18 [21] | 79.43% | 56.17% | 34.86% | 28.23% | 23.45% |
| SCN [15] | 80.89% | 60.21% | 44.97% | 37.79% | 30.11% |
| SDNet-18 w/ $\lambda = 0.1$ | 81.78% | 63.50% | 43.86% | 35.84% | 27.92% |
| SDNet-18 w/ adaptive $\lambda$ | 84.76% | 74.87% | 61.38% | 54.77% | 48.84% |
| $\lambda$ from linear fitting | 0.49 | 0.60 | 0.75 | 0.84 | 0.94 |

Table 3: Classification result under varying corruption types (averaged over all severity levels for each type) on CIFAR-10-C and ImageNet-C with $\lambda = 0.1$ (i.e., same as training) and with an adaptive $\lambda$ computed from Algorithm 1.

| | CIFAR-10-C | | | | ImageNet-C | | |
|---|---|---|---|---|---|---|---|
| Noise Type | Gaussian | Shot | Speckle | Impulse | Gaussian | Shot | Impulse |
| ResNet-18 [21] | 44.43% | 57.88% | 62.16% | 51.72% | 22.73% | 21.78% | 17.38% |
| SCN [15] | 50.79% | 62.97% | 67.45% | 54.19% | - | - | - |
| SDNet-18 w/ $\lambda = 0.1$ | 50.58% | 63.29% | 67.11% | 54.13% | 24.98% | 23.97% | 19.12% |
| SDNet-18 w/ adaptive $\lambda$ | 64.92% | 71.13% | 71.42% | 57.48% | 29.16% | 27.59% | 22.01% |

that sparse modeling enables us to effectively handle various types of additive noises in test data very easily with the procedure in Algorithm 1.

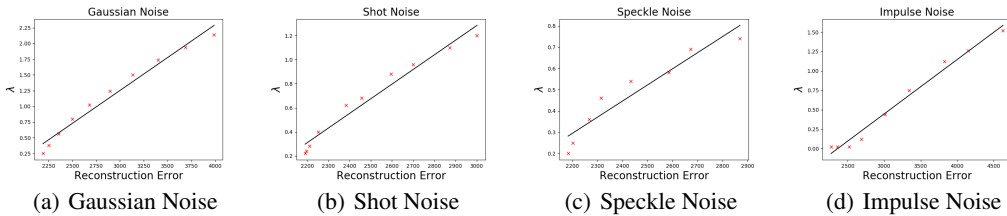

(a) Gaussian Noise     (b) Shot Noise     (c) Speckle Noise     (d) Impulse Noise

Figure 3: Relation between the optimal choice of $\lambda$ and the magnitude of residual in CSC-layers (i.e., $\{\lambda_c, r_c\}_{c \in \mathcal{C}}$ in Algorithm 1, drawn as red crosses) as well as the linear fitting (i.e., $\lambda = \lambda(c)$ in Algorithm 1, drawn as black lines) for four corruption types $\mathtt{T} \in \{\text{Gaussian, Shot, Speckle, Impulse}\}$.

Finally, in Figure 3 we plot the relationship between the optimal choice of $\lambda$ (in y-axis) and the magnitude of residual in CSC-layers (in x-axis) learned from training data according to Algorithm 1. We also plot the linear fitting of such a relation, which can be seen to provide a good quality approximation for each of the four corruption types. On the other hand, we note that the fitted linear relations differ across different types of corruption. Hence, it is important that the relationship is estimated for each corruption type separately.

### 4.3 Handling Adversarial Perturbations

We show that our method also exhibits robustness to adversarial perturbations. In this experiment, we generate adversarial perturbations on the CIFAR-10 test dataset using PGD attack on our SDNet (with $\lambda = 0.1$), with $L_\infty$ norm of the perturbation being $\epsilon = 8/255$ and $L_2$ norm of the perturbation being $\epsilon = 0.5$, respectively. The robust accuracy of our method is reported in Table 4 and is compared with that of ResNet-18. We can see that while SDNet does not perform much better than ResNet with $\lambda = 0.1$, we may tune the parameter $\lambda$ to drastically improve the robust accuracy.

### 4.4 Analysis

**Effect of number of iterations.** Recall from Sec. 3.1 that forward propagation through a CSC-layer is performed by running a few iterative steps of the FISTA algorithm. Here we provide a study on how the number of iterations affects model performance on ImageNet and ImageNet-C using SDNet-18. The results are shown in Table 5. With the increasing number of FISTA iterations, the model performance on both natural accuracy and robust accuracy improves.

Table 4: Robust accuracy on CIFAR-10 with adversarial perturbation using PGD attack.

| Model | Robust Accuracy ($L_\infty = 8/255$) | Robust Accuracy ($L_2 = 0.5$) |
|---|---|---|
| ResNet-18 [21] | 0.01% | 29.47% |
| SDNet-18 w/ $\lambda = 0.1$ | 0.11% | 29.95% |
| SDNet-18 (After tuning $\lambda$) | 35.18% | 62.80% |

Table 5: Effect of number of FISTA iteration on natural and robust accuracy (evaluated with ImageNet-C) for SDNet-18 trained on ImageNet.

| # of FISTA Iterations | Natural Accuracy | Gaussian | Shot | Impulse |
|---|---|---|---|---|
| 2 | 69.47% | 29.16% | 27.59% | 22.01% |
| 4 | 69.51% | 29.69% | 28.15% | 24.15% |
| 8 | 69.79% | 30.91% | 29.87% | 25.69% |

**Replacing all convolution layer by the CSC-layer.** We also train a version of SDNet obtained from replacing all convolution layers of a ResNet with the CSC-layer (as opposed to only the first convolution layer), and refer to such a model as SDNet-18-All and SDNet-34-All. On ImageNet we observe that SDNet-18-All and SDNet-34-All obtain 69.37% and 72.54% Top-1 accuracy, respectively. Comparing such results with those of SDNet-18/34 reported in Table 1, we see that the performance is not significantly affected by replacing more convolution layers with CSC-layers (see Table D.1 in Appendix for more results). Moreover, SDNet-18/34-All enables us to develop a visualization technique as described in the Appendix B.

## 5   Conclusion and Discussion

This paper revisits the classical sparse modeling and provides a simple way of using it to guide the design of interpretable deep networks. Despite multiple prior attempts, our work is the first to demonstrate that such a design can produce performance (in terms of accuracy, model size, and memory) that is on par with standard ConvNets on modern image datasets such as ImageNet. The success in combining sparse modeling with deep learning provides a means of borrowing and utilizing the rich results in the well-developed field of sparse modeling [54, 55] for network design and analysis. While it is not the purpose of this work to fully explore all potentials, our experiments already demonstrate clear advantages of so-designed networks in handling various forms of data corruption. Looking forward, other fundamental principles, algorithms, and techniques in sparse modeling may be introduced to further enhance the capability of our presented framework. Along this line, we provide some preliminary evidence on how sparse modeling enables interpretability in the Appendix, and leave further study to future work.

## Acknowledgments and Disclosure of Funding

Zhihui Zhu acknowledges support from NSF grants CCF-2240708. Shao-Lun Huang acknowledges support from Shenzhen Science and Technology Program under Grant KQTD20170810150821146, National Key R&D Program of China under Grant 2021YFA0715202 and High-end Foreign Expert Talent Introduction Plan under Grant G2021032013L. Yi Ma acknowledges support from ONR grants N00014-20-1-2002 and N00014-22-1-2102, the joint Simons Foundation-NSF DMS grant #2031899, as well as partial support from Berkeley FHL Vive Center for Enhanced Reality and Berkeley Center for Augmented Cognition, Tsinghua-Berkeley Shenzhen Institute (TBSI) Research Fund, and Berkeley AI Research (BAIR).

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
