# Revisiting Sparse Convolutional Model for Visual Recognition
## – *Supplementary Material* –

**Xili Dai**[1] [*]    **Mingyang Li**[2] [*]    **Pengyuan Zhai**[3]    **Shengbang Tong**[4]    **Xingjian Gao**[4]
**Shao-Lun Huang**[2]    **Zhihui Zhu**[5]    **Chong You**[4]    **Yi Ma**[2,4]
[1]The Hong Kong University of Science and Technology (Guangzhou)
[2]Tsinghua-Berkeley Shenzhen Institute (TBSI), Tsinghua University
[3] Harvard University    [4] University of California, Berkeley    [5] Ohio State University

# Appendices

We provide the implementation details for the CSC-layers in Appendix A, followed by a discussion on how sparse modeling enables us to easily visualize the feature maps in the intermediate layers of a convolutional neural network in Appendix B.

## A    Implementation Details for CSC-Layers

**Forward propagation.** Forward propagation of the sparse coding layer is carried out by solving the optimization problem in (4). In this paper, we adopt the fast iterative shrinkage thresholding algorithm (FISTA) [1]. Starting with an arbitrarily initialized $z$ (we used $z^{[0]} = \mathbf{0}$), FISTA operates by introducing an auxiliary variable $y$ initialized as $y^{[1]} = z^{[0]}$, as well as a scalar $m$ initialized as $m_1 = 1$, and carrying out iteratively the following steps for $\ell \geq 1$:

$$z^{[\ell]} = \mathcal{T}_{\lambda t}\big(y^{[\ell]} + t\mathcal{A}^*(x - \mathcal{A}(y^{[\ell]}))\big),$$

$$m_{\ell+1} = \frac{1 + \sqrt{1 + 4m_\ell^2}}{2}, \tag{A.1}$$

$$y^{[\ell+1]} = z^{[\ell]} + \frac{m_k - 1}{m_{k+1}}(z^{[\ell]} - z^{[\ell-1]})$$

In above, $\mathcal{A}^*(\cdot)$ is the adjoint operator of $\mathcal{A}(\cdot)$ and it is defined as

$$\mathcal{A}^* x := \sum_{m=1}^{M} \big(\boldsymbol{\alpha}_{m1} * \boldsymbol{\xi}_m, \ldots, \boldsymbol{\alpha}_{mC} * \boldsymbol{\xi}_m\big) \in \mathbb{R}^{C \times H \times W}. \tag{A.2}$$

for any $x = (\boldsymbol{\xi}_1, \ldots, \boldsymbol{\xi}_M) \in \mathbb{R}^{M \times H \times W}$, where each $\boldsymbol{\xi}_c \in \mathbb{R}^{H \times W}$.

The FISTA iteration in (A.1) automatically gives rise to a nonlinear operator $\mathcal{T}_{\lambda t}$, which denotes a shrinkage thresholding operator that is applied entry-wise to the input variable. In particular, shrinkage thresholding for a scalar $\eta \in \mathbb{R}$ is defined as

$$\mathcal{T}_{\lambda t}(\eta) \doteq \max(|\eta| - \lambda t, 0) \cdot \text{sgn}(\eta). \tag{A.3}$$

Finally, the parameter $t$ is the step size for FISTA. The iterations (A.1) converge to a solution to (4) as long as the step size $t$ is smaller than the inverse of the dominant eigenvalue of the operator

$\mathcal{A}^\top(\mathcal{A}(\cdot))$. In our implementation, we estimate this dominant eigenvalue by the power iteration, which iteratively carries out the following calculation to estimate the dominant eigenvector:

$$\boldsymbol{v}^{[k+1]} \leftarrow \frac{\tilde{\boldsymbol{v}}^{[k+1]}}{\|\tilde{\boldsymbol{v}}^{[k+1]}\|_2}, \text{ where } \tilde{\boldsymbol{v}}^{[k+1]} = \mathcal{A}^\top(\mathcal{A}(\boldsymbol{v}^{[k]})). \tag{A.4}$$

The iteration is terminated when the update $\boldsymbol{v}^{[k+1]} - \boldsymbol{v}^{[k]}$ has an entry-wise $\ell_2$-norm smaller than a predefined threshold (we use $\sqrt{10^{-5}}$) or when the maximum number of iterations (we use 50) is reached. Assuming that the iteration carries out for $K$ times, the estimated dominant eigenvalue is given by

$$\lambda_K = \langle \boldsymbol{v}^{[K]}, \mathcal{A}^* \mathcal{A} \boldsymbol{v}^{[K]} \rangle. \tag{A.5}$$

We may then set $t$ to be a value that is smaller than $1.0/\lambda_K$ to guarantee the convergence of ISTA. In our experiment we set $t = 0.9/\lambda_K$. Note that the dominant eigenvalue for $\mathcal{A}^\top(\mathcal{A}(\cdot))$ changes after each update on the layer parameters $\boldsymbol{A}$. Hence, the calculation of $\lambda_K$ is performed after each parameter update during training. Once training is completed, $\lambda_K$ can be fixed during testing and therefore does not incur additional inference time.

**Backward propagation.** A benefit of adopting the FISTA iteration for forward propagation in the sparse coding layer is that it naturally leads to an optimization-driven network [2], a network architecture that is constructed from an unrolled optimization algorithm, for which backward propagation can be carried out by auto-differentiation.

We note that there are many other algorithms for solving the sparse coding and convolutional sparse coding [3] problems, which may lead to more efficient implementation of the forward propagation. Moreover, there are also other algorithms for performing the backward propagation [4, 5] by leveraging the fact that the sparse coding problem (4) is convex. We choose FISTA because it has a very simple implementation for both forward and backward propagation. We leave the acceleration of our approach with other solvers as future work.

**Implementation.** Implementation of the sparse coding layer requires carrying out the computation of the operators $\mathcal{A}(\cdot)$ and $\mathcal{A}^*(\cdot)$. Both of these two operators can be easily implemented by many modern deep learning packages, as noted in [6]. In PyTorch, for example, $\mathcal{A}(\boldsymbol{z})$ can be implemented by the convolutional function as follows:

$$\texttt{torch.nn.functional.conv2d(input, weight)}, \tag{A.6}$$

where `input` is set to $\boldsymbol{z}$ (after inserting a batch dimension) and `weight` is set to $\boldsymbol{A}$. Similarly, $\mathcal{A}^*\boldsymbol{x}$ can be easily implemented by replacing the `conv2d` function in (A.6) with the `conv_transposed2d` function.

## B   Visualization of Feature Maps with CSC-layers

Visualization of the feature maps in the intermediate layers of a deep neural network is an important aspect toward establishing the interpretability of deep learning models. Here we illustrate how sparse modeling with CSC-layers enables us to very easily obtain such a visualization in ways that cannot be accomplished with standard convolutional layers.

Before we introduce our visualization method, we mention that existing convolutional neural network visualization methods primarily include those based on 1) synthesizing an input image pattern that maximally activates one [7, 8] or multiple [9] CNN neurons and 2) mapping the intermediate features into visually perceptible signals by training reversed convolutional filters [10] or by using the transposed versions of the (forward-pass) convolutional filters along with approximated unpooling operations [11]. These methods have the shortcomings such as requiring extra training steps [7, 8, 9, 10], visualizing only in the neuron level and lacking layer-level perspectives [7, 8, 11], and relying on numerically approximated inverse operators [11, 10].

In contrast to previous methods, our method enables an analytical reconstruction of the layer input from the layer output (feature map) by simply applying the learned convolutional dictionary on the feature map without additional training, visualization level restrictions, or any approximated inverse operators. As we explain next, this is made possible by sparse modeling with CSC-layers.

### B.1 Method

Recall that each CSC-layer solves an optimization problem (4) where the layer input is reconstructed by the layer output with layer parameter being the dictionary. Specifically, let $\mathcal{A}_l$ be the (learned) convolutional dictionary and $z_{l-1}$, $z_l$ be the input and output feature maps respectively, at layer $\ell$. In CSC-layers, $z_l$ is the optimal solution to (4), replicated here for convenience:

$$z_l = \arg\min_z \lambda \|z\|_1 + \frac{1}{2}\|z_{l-1} - \mathcal{A}_l(z)\|_2^2. \tag{B.1}$$

Given a feature map $z_l$ at the output of a layer $\ell$, one may obtain a reconstructed layer input $\tilde{z}_{l-1}$ from the convolution operation, i.e., $\tilde{z}_{l-1} \doteq \mathcal{A}_l(z_l)$. In a deep model with multiple CSC-layers, one may apply this reconstruction step recursively from the $\ell$-th layer to the first layer, to obtain an reconstructed input image $\tilde{x}$:

$$\begin{aligned} \tilde{z}_{\ell-1} &\doteq \mathcal{A}_n(z_\ell) \\ \tilde{z}_{i-1} &\doteq \mathcal{A}_i(\tilde{z}_i), \quad i = 2, \dots, \ell - 1 \\ \tilde{x} &\doteq \mathcal{A}_1(z_1). \end{aligned} \tag{B.2}$$

To express this more compactly, we have

$$\tilde{x} \approx \mathcal{A}_1(\mathcal{A}_2(\dots(\mathcal{A}_{\ell-1}(\mathcal{A}_\ell(z_\ell))))). \tag{B.3}$$

The output $\tilde{x}$, which is in the input image space, can then be visualized as an image. Note that the procedure in (B.3) for obtaining $\tilde{x}$ is analytical and thus does not require numerical approximation or additional training, as opposed to [11, 10] and [7, 8, 9, 10].

### B.2 Results

We apply our visualization method described above to the SDNet-18 trained on ImageNet (see Sec. 4) to visualize the feature maps associated with five different images. The results are provided in Figure B.1. It can be observed that the shallow layers (e.g., layer 1 - 5) capture rich details of the input image. On the other hand, feature maps at deeper layers become more blurry and only capture a rough contour of the contents in the corresponding input image. This is suggesting that the layers of SDNet-18 progressively remove some of the unrelated details from the network input.

## C  Visualizing the Learned Dictionary of SDNet-18

In Figure C.1, we provide a visualization of the learned dictionary in the first layer of SDNet-18 trained on ImageNet. The dimension of the dictionary in first layer of SDNet-18 is $64 \times 3 \times 7 \times 7$, which is treated as 64 small patches of shape $7 \times 7$ with RGB channels that are arranged into a $8 \times 8$ grid.

## D  More Experimental Results

### D.1  Sparsity of CSC-layer Feature Map

To evaluate the sparsity of our CSC-layer output feature map, we generate a histogram of the values at the output of CSC-layer in SDNet-18 for 10,000 CIFAR-10 test images and report the results in Figure D.1. The result shows our CSC-layer feature maps are highly sparse. The zero value accounts for 52% of whole CSC-layer feature maps while common convolution layer outputs dense feature maps.

### D.2  Comparison of SDNet-18 and SDNet-18-All

Table D.1 shows the comparison of SDNet-18/34 and SDNet-18/34-All on CIFAR-10, CIFAR-100 and ImageNet datasets, including accuracy, model complexity and speed. SDNet-18-All means all convolution layers are replaced with CSC-layer. And the number of FISTA iteration is set to 2 for all CSC-layers. Both models have high accuracy performance while SDNet-18 is significantly faster.

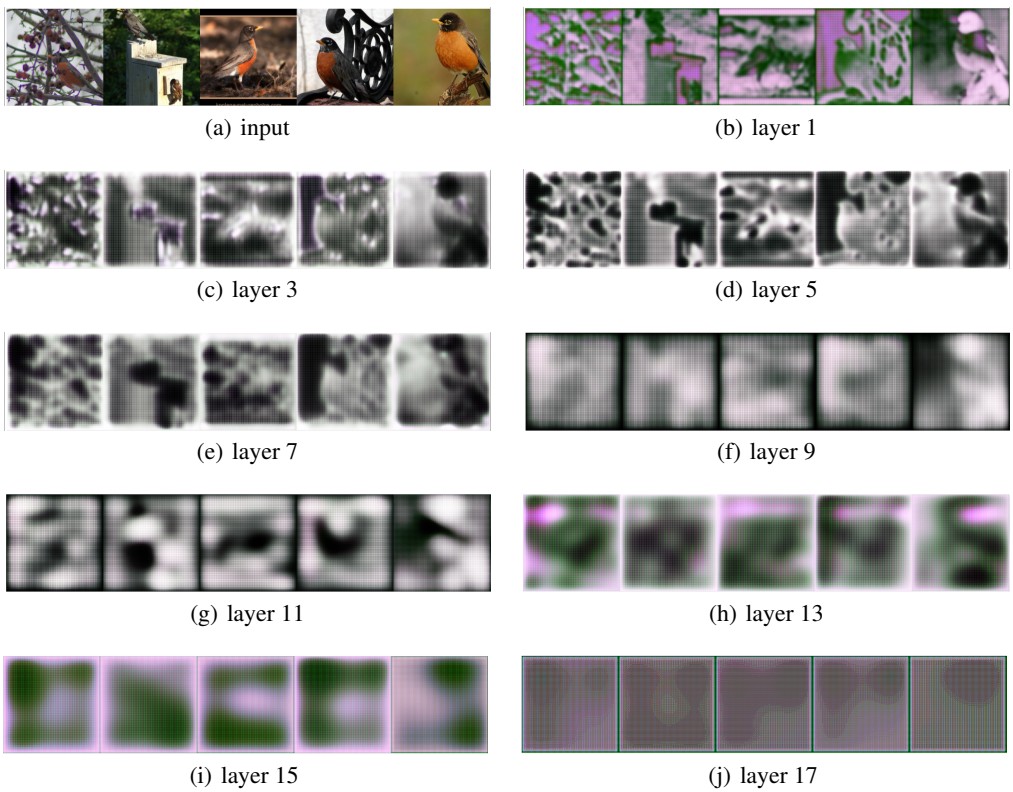

(a) input

(b) layer 1

(c) layer 3

(d) layer 5

(e) layer 7

(f) layer 9

(g) layer 11

(h) layer 13

(i) layer 15

(j) layer 17

Figure B.1: Visualization of feature maps for 5 images at selected layers of a SDNet-18 trained on ImageNet.

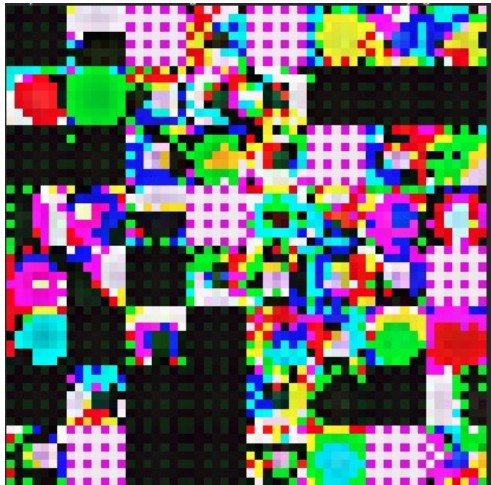

Figure C.1: Visualization of the learned dictionary of first layer of SDNet-18-All trained on ImageNet.

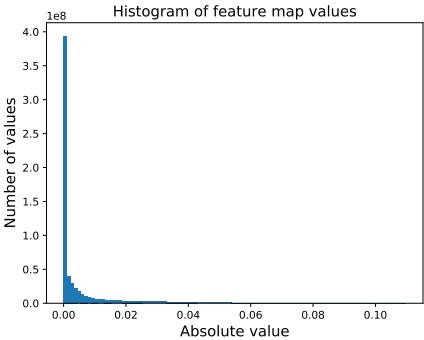

Figure D.1: Histogram of CSC-layer feature map values in a trained SDNet-18 using all CIFAR-10 test images.

Table D.1: Comparison of SDNet-18 and SDNet-18-All.

|  | Model Size | Top-1 Acc | Memory | Speed |
|---|---|---|---|---|
| **CIFAR-10** | | | | |
| SDNet-18 | 11.2M | 95.20% | 1.2 GB | 1500 n/s |
| SDNet-18-all | 11.2M | 95.18% | 4.1 GB | 400 n/s |
| **CIFAR-100** | | | | |
| SDNet-18 | 11.3M | 78.31% | 1.2 GB | 1500 n/s |
| SDNet-18-all | 11.3M | 78.16% | 4.1 GB | 400 n/s |
| **ImageNet** | | | | |
| SDNet-18 | 11.7M | 69.47% | 37.6 GB | 1800 n/s |
| SDNet-18-all | 11.7M | 69.37% | 121.9 GB | 490 n/s |
| SDNet-34 | 21.5M | 72.67% | 46.4 GB | 1200 n/s |
| SDNet-34-all | 21.5M | 72.54% | 157.7 GB | 300 n/s |