# OpenReview forum: "Revisiting Sparse Convolutional Model for Visual Recognition"
_NeurIPS.cc/2022/Conference — NeurIPS 2022 Accept_

### Official Review · Reviewer_WxVJ · 2022-07-11

**Rating:** 4
**Confidence:** 4
**Soundness:** 2 fair
**Presentation:** 3 good
**Contribution:** 2 fair

**Summary:**

The paper proposes to replace the traditional convolutional layers with convolutional sparse coding (CSC) layers, claiming that such substitution adds interpretability and robustness to neural networks.

**Questions:**

I have three main concerns about the claims in the paper.

**1. Interpretability**. The traditional convolutional layer performs a forward computation (the output is a linear combination of the inputs). In contrast, the convolutional sparse coding (CSC) layer performs a backward computation (the input is a linear combination of the outputs). It is not apparent why a backward computation is more interpretable than a forward one. In my opinion, it is not an individual layer that makes a neural network hard to interpret but the stack of these layers. While convolution layer and convolutional sparse coding are easy to interpret individually, using them in deep networks (with nonlinearities, normalization, etc.) is not.

**2. Robustness**. The explanation of why the CSC layer is more robust is insufficient --- no math derivation is used to explain this concept. From the writing, it seems the Lipschitz constant decreases if the regularizer increases. In this case, the comparison to standard convolutional networks is not fair. Maybe the authors would also like to enforce the Lipschitzness of traditional convolutional networks (e.g., following https://arxiv.org/abs/1804.04368).

**3. Computational Complexities**. It looks to me that the proposed layer is quite expensive. In the experiment, only one layer in ResNet is replaced by the proposed layer, and only two iterations are used in unrolling. And this already decreases the speed from 1000 to 900. I think a more comprehensive study on the relationship between accuracy, complexity, and iterations is needed when all layers are replaced.

**Limitations:**

Not applicable.

**Strengths And Weaknesses:**

The writing is generally good and easy to follow; however, the technical claims in the paper are not well supported by empirical evidence.

---

> ### Author Response · Authors · 2022-08-02
> **Response to Reviewer WxVJ**
>
> We thank the reviewer's comments but disagree with the suggested rating: our paper is both technically sound and experimental evaluation is rather thorough and convincing. We clarify some of your concerns or possible misunderstandings below, which hopefully can change your opinion about our paper:
>
> >Q1: Interpretability. The traditional convolutional layer performs a forward computation (the output is a linear combination of the inputs). In contrast, the convolutional sparse coding (CSC) layer performs a backward computation (the input is a linear combination of the outputs). It is not apparent why a backward computation is more interpretable than a forward one. In my opinion, it is not an individual layer that makes a neural network hard to interpret but the stack of these layers. While convolution layer and convolutional sparse coding are easy to interpret individually, using them in deep networks (with nonlinearities, normalization, etc.) is not.
>
> A: We note that we have never claimed that the CSC layer offers interpretability of the entire deep neural network. Rather, our claim is that the CSC layer itself offers interpretability, in the sense that it models the input as a sparse linear combination of a (learned) convolutional dictionary. Importantly, such an interpretation allows us to design a new technique for improving network robustness by leveraging the stable recovery properties of the sparse modeling, as well as a means of visualizing feature maps due to the fact that a CSC layer is (locally) generative and can naturally reproduce the input from its output. Notably, standard forward convolution layers do not provide such means of obtaining robustness and for feature visualization (hence interpretation).
>
> We realized that our usage of “interpretability” in line 55 may have caused confusion for the reviewer. Hence, we have updated our writing in the revised version to further clarify.
>
> >Q2: Robustness. The explanation of why the CSC layer is more robust is insufficient --- no math derivation is used to explain this concept. From the writing, it seems the Lipschitz constant decreases if the regularizer increases. In this case, the comparison to standard convolutional networks is not fair. Maybe the authors would also like to enforce the Lipschitzness of traditional convolutional networks (e.g., following https://arxiv.org/abs/1804.04368).
>
> A: The fact that CSC is robust to input perturbation is well-established in previous work [42, Theorem 19] as we have discussed in Sec. 3.3. In the revised version, we have explicitly included a restatement of such results with rigorous mathematical characterization to more clearly explain the concept.
>
> Regarding Lipschitz constant: While we have never computed Lipschitz constant for our proposed SDNet, we agree with the reviewer that our method should have a smaller Lipschitz constant as it provides a stable recovery for the input. However, unlike commonly used techniques for improving Lipschitzness properties that usually improves robustness at the cost of a lower performance on clean data, our technique does not affect the performance on clean data at all.
>
> >Q3: Computational Complexities. It looks to me that the proposed layer is quite expensive. In the experiment, only one layer in ResNet is replaced by the proposed layer, and only two iterations are used in unrolling. And this already decreases the speed from 1000 to 900. I think a more comprehensive study on the relationship between accuracy, complexity, and iterations is needed when all layers are replaced.
>
> A: The following table shows the comparison of SDNet-18 and SDNet-18-All on accuracy, complexity. SDNet-18-All means all convolution layers are replaced with CSC-layer. And the number of FISTA iteration is two for all CSC-layers, hence the complexity is only twice. In the new supplementary material, we have also conducted ablation studies on the number of iterations on ImageNet, see Table D.1.
>
> |                                 | Model Size     |     Top-1 Acc    |      Memory     |       Speed|
> |----------------------|-------------------|------------------|-----------------|------------|
> SDNet-18                |     11.2M          |      95.20%       |      1.2GB        |  1500 n/s |
> SDNet-18-all           |     11.2M          |      95.18%       |       2.5GB       |   720  n/s |

---

> ### Author Response · Authors · 2022-08-07
> **Further discussion with the Reviewer**
>
> Dear reviewer WxVJ:
>
> We thank you for the precious review time and valuable comments. We have provided corresponding responses and results, which we believe have covered your concerns. We hope to further discuss with you whether or not your concerns have been addressed. Please let us know if you still have any unclear parts of our work.

---

### Official Review · Reviewer_EVjq · 2022-07-11

**Rating:** 6
**Confidence:** 4
**Soundness:** 3 good
**Presentation:** 3 good
**Contribution:** 3 good

**Summary:**

This paper proposes an approach to incorporate convolutional sparse coding into deep neural networks. In particular, a convolutional layer of a ResNet is replaced with an implicit layer, referred to as convolutional sparse coding layer (CSC-layer), which outputs a sparse representation (a feature map) of its input given a (learned) dictionary of convolutional filters. The sparse representation for an input is computed using the FISTA algorithm. The experiments show that ResNet models whose first convolutional layer is replaced by such a CSC-layer can achieve similar or better classification performance on the CIFAR-10, CIFAR-100, and ImageNet datasets. Additionally, experiments suggest that ResNet models with a CSC-layer are more robust to different kinds of additive noise applied to the input when compared to a regular ResNet model.

**Questions:**

1. Have the authors considered visualizing the learned sparse dictionary convolutional kernels common in related literature (eg. [1])? I believe this would help with interpretability and understanding what the dictionary of convolutional kernels encodes.

2. Typically, the FISTA algorithm requires hundreds of iterations to converge so my expectation is that the reconstructions $\tilde{x} = \mathcal{A}(z_*)$ with only 2 iterations are not high fidelity (e.g., terms of PSNR). This is supported by the visualization in Appendix B2 which shows that feature maps only encode contours or high-level information about the input. The authors mention that increasing the number of FISTA iterations can boost the classification performance a bit. Have the authors’ studied how increasing the number of FISTA iterations affects the model’s robustness to noise or can they provide intuition about it?

3. My understanding is that only the first convolutional layer of ResNet-18 and ResNet-34 (the one closest to the input) is replaced by a CSC-layer. Is this correct or does “the first convolutional layers” (line 235) refer to the first convolutional layer of each ResNet block?

4. How is the value of $\lambda = 0.1$ used during training selected? What is the size of $C$ used in experiments, i.e. the number of sparse feature maps in $z$ (line 125)? How sparse on average are the feature maps output by FISTA when only 2 iterations are used with regularization coefficient $\lambda = 0.1$?

5. What magnitudes do levels 0-6 in Figure 2 correspond to for each type of noise? E.g. for Gaussian noise, what levels of noise are considered? Same for Tables 2 and 3.

6. Have the authors studied how robust SD SCN model (from [3]) is to additive noise? Would be insightful to add it as a baseline in Tables 2 and 3.

7. Are the authors planning to provide an implementation of the proposed framework?

8. Nitpicks
- Figures 2 and 3 would be easier to read if the font size in is bigger.

[1] Kavukcuoglu, K., Sermanet, P., Boureau, Y.L., Gregor, K., Mathieu, M. and LeCun, Y., 2010. Learning convolutional feature hierarchies for visual recognition. Advances in neural information processing systems, 23.

[2] Beck, A. and Teboulle, M., 2009. A fast iterative shrinkage-thresholding algorithm for linear inverse problems. SIAM journal on imaging sciences, 2(1), pp.183-202.

[3] Sun, X., Nasrabadi, N.M. and Tran, T.D., 2018, October. Supervised deep sparse coding networks. In 2018 25th IEEE International Conference on Image Processing (ICIP) (pp. 346-350). IEEE.


**Limitations:**

The authors are upfront about the limitations of higher memory and inference costs of unrolling FISTA for a higher number of iterations. They could elaborate on the potential negative societal impacts, for example, the fact that the model can propagate any bias present in the dataset.


**Strengths And Weaknesses:**

Strengths

- The proposed method which combines sparse coding with deep neural networks is scalable to datasets such as ImageNet.

- With only 2 iterations of FISTA, the proposed modification of the ResNet model is more robust to additive noise in the input than the vanilla ResNet architecture when evaluated on the task of classification.

- The paper is organized and written clearly.

---

> ### Author Response · Authors · 2022-08-02
> **Response to Reviewer EVjq**
>
> We thank the reviewer for the careful reading of our manuscript and the constructive remarks. Here we reply to those specific questions.
>
> >Q1: Have the authors considered visualizing the learned sparse dictionary convolutional kernels common in related literature ? I believe this would help with interpretability and understanding what the dictionary of convolutional kernels encodes.
>
> A: Thank you for the comment. We visualize the learned dictionary and post it in the appendix (Figure C.1) of the revised version.
>
> >Q2: Typically, the FISTA algorithm requires hundreds of iterations to converge so my expectation is that the reconstructions x=Az  with only 2 iterations are not high fidelity (e.g., terms of PSNR). This is supported by the visualization in Appendix B2 which shows that feature maps only encode contours or high-level information about the input. The authors mention that increasing the number of FISTA iterations can boost the classification performance a bit. Have the authors’ studied how increasing the number of FISTA iterations affects the model’s robustness to noise or can they provide intuition about it?
>
> A: Thank you for the comments. The following table shows how the number of FISTA iterations affects the model’s robustness to noise. The model is trained on the ImageNet dataset. The “natural accuracy” column is the accuracy tested on the validation set of ImageNet, the columns “Gaussian”, “Shot”, and “Impulse” are three different noises from ImageNet-C. We report the top-1 accuracy results with adaptive lambda. While using more iterations slightly increases the model performance on both natural accuracy and robust accuracy.
>
> |# of FISTA iterations       |natural accuracy  | Gaussian  | Shot       |  Impulse|
> |----------------------------|---------------------|-------------|-----------|-----------|
> |2                                     | 69.47%                |  29.16%   |  27.59% |  22.01%|
> |4                                     | 69.51%                |  29.69%   |  28.15% |  24.15%|
> |8                                     | 69.79%                |  30.91%   |  29.87% |  26.69%|
>
> >Q3: My understanding is that only the first convolutional layer of ResNet-18 and ResNet-34 (the one closest to the input) is replaced by a CSC-layer. Is this correct or does “the first convolutional layers” (line 235) refer to the first convolutional layer of each ResNet block?
>
> A: Yes, only the first convolutional layer of ResNet-18 and ResNet-34 (the one closest to the input) is replaced by a CSC-layer.
>
> >Q4: How is the value of lmdb=0.1 used during training selected? What is the size of C used in experiments, i.e. the number of sparse feature maps in  (line 125)? How sparse on average are the feature maps output by FISTA when only 2 iterations are used with regularization coefficient?
>
> A: The value of $\lambda$ was selected based on grid search and the one corresponding to the best test accuracy was chosen. The number of sparse feature maps is the same as the channel number of ResNet in each layer, which are 3 -> 64 -> 128 ->256 -> 512 as in each block of ResNet18/34. We also test the sparsity of the feature map on all 10000 CIFAR-10 test samples and find that 52% values are exactly 0, while the feature map of the convolutional layer in ResNet is dense. The histogram of the feature map absolute values is shown in the appendix (Figure D.1) of the revised version.
>
> >Q5: What magnitudes do levels 0-6 in Figure 2 correspond to for each type of noise? E.g. for Gaussian noise, what levels of noise are considered? Same for Tables 2 and 3.
>
> A: In our experiments, we use the CIFAR-C and ImageNet-C data. The noises are added to the clean data with pixel values in the range of [0, 1]. The specific noise parameters from severity level 1-5 are as follows. For the gaussian noise, the standard deviation is 0.08, 0.12, 0.18, 0.26, 0.38. For the shot noise, the value of parameters are 60, 25, 12, 5, 3. For the impulse noise, the amount of s&p impulses are 0.03, 0.06, 0.09, 0.17, 0.27. For the speckle noise, the standard deviation of the gaussian multiplier is 0.15, 0.2, 0.35, 0.45, 0.6. For the detailed implementation, please check the code of [1].
>
> [1] Hendrycks D, et al. Benchmarking neural network robustness to common corruptions and perturbations[J]. arXiv:1903.12261, 2019.
>
> >Q6: Have the authors studied how robust SD SCN model is to additive noise? Would be insightful to add it as a baseline in Tables 2 and 3.
>
> A: We test the robustness of SCN on CIFAR10-C and show its results in Table 2,3 of the revised version. The results are very close to our SDNet18 w/ $\lambda$=0.1
>
> >Q7: Are the authors planning to provide an implementation of the proposed framework?
>
> A: Thank you for the suggestion, we will release the code.
>
> >Q8: Nitpicks. Figures 2 and 3 would be easier to read if the font size is bigger.
>
> A: Thanks for pointing out this. We have fixed it in the revised version.

---

> > ### Comment · Reviewer_EVjq · 2022-08-05
> > **Thank you for your response!**
> >
> > I thank the authors for the detailed response and additional analysis and experiments! Please find below some follow-up comments.
> >
> > Additionally, I’d also like to apologize that I had prepared a “Room for Improvement” section in my original review (please find below) but when transferring to the Open Review platform missed to include it. However, my questions reflected most of the concerns raised there so my omission doesn’t influence the current discussion.
> >
> > **Room for improvement**
> > - I believe it would be helpful to visualize the learned sparse dictionary convolutional kernels which is common in other related work (e.g. [1]) and would provide insight into the signals that the dictionary can encode.
> > - The classification experiments are missing confidence intervals. I believe the results would be enhanced if error bars are included.
> > - I pose a few clarifications on the choice of hyperparameters and experimental setup that would enhance interpreting the results in the paper.
> > - To promote reproducibility, it would be helpful for the authors to provide an open-source implementation of the proposed method.
> >
> > **Q1**: Visualizing the dictionary elements of the first ResNet convolutional layer 64 x 3 x 7 x 7.
> >
> > Most of the convolutional filters look noisy and unlike the edge, orientation, and center-surround detectors typically found in sparse dictionaries. I wonder whether this is due to the training not being fully converged or maybe the fact that only 2 iterations of FISTA are used. Also, it would be helpful to add a border between the different filters in the figure in order to separate them.
> >
> > **Q2**: Thank you for this additional result on how the number of FISTA iterations affects the model’s robustness to noise.
> >
> > **Q3**: Thank you for clarifying  that only the bottom layer of ResNet is replaced by a CSC-layer.
> >
> > **Q4**: Thank you for specifying how the value of lambda was selected and for providing a histogram with the distribution of feature map values.
> >
> > **Q5**: Thank you for clarifying the magnitudes of levels 0-6 in Figure 2.
> >
> > **Q6**: Thank you for providing the additional baseline.
> >
> > **Q7**: Thank you for planning to release the implementation.

---

> > > ### Author Response · Authors · 2022-08-09
> > > **Thanks for your recognition and response!**
> > >
> > > Dear reviewer EVjq,
> > >
> > > For Q1, it might be caused by the number of FISTA iterations. We will conduct more ablation studies and visualization in the future version.
> > > Also, thanks for the suggestion about the visualization, we will add a border between different filters to make better visualization.

---

### Official Review · Reviewer_nUHw · 2022-07-13

**Rating:** 7
**Confidence:** 3
**Soundness:** 3 good
**Presentation:** 3 good
**Contribution:** 3 good

**Summary:**

This paper proposed the convolutional sparse coding layer (CSC-layer) to obtain performance comparable to standard ConvNets with better interpretability and stability.  By conducting extensive experiments on CIFAR-10, CIFAR-100, and ImageNet, the model with CSC-layer has shown to achieve comparable performance as standard ConvNets and be more robust to data perturbation than the standard one.

**Questions:**

1. In Table 1, the model size of SDNet-18 and SDNNet-34 on CIFAR-100 are much smaller than on CIFAR-10, which seems wired.
2. With similar performance, the proposed method is much faster than its baselines. In Table 1, the proposed SDNet only replaces the first convolutional layer with CSC-layer while SCN is a multilayer sparse coding network. For a fair comparison, this paper may compare the time and memory consumption of a single sparse coding layer between those methods.
3. Each layer of CSC-layer of SDNet-18 and SDNet34 needs unrolling two iterations of FISTA and more iterations will only slightly improve the performance. As SDNet-18 and SDNet-34 have only one CSC-layer for the input images, I’m curious whether it is this low dimension (3 channels) of input that make two iterations sufficient. On SDNet-18-All and SDNet-34-All, could you list the dimension of the input and output of each CSC-layers and their corresponding iterations used?
4. What is the cost to find the optimal $\lambda$ and calculate the residual in Algorithm 1?
5. I would like to confirm whether it is $z$ of a CSC-layer that will be the input of the next layers.
6. Does FISTA algorithm always randomly initialize $z$ for any CSC-layer in any iteration during the training? If it is, is it possible to initialize it with the previous leaned values for an image when the model sees it again? This may reduce the number of FISTA iterations.


**Limitations:**

This paper introduced an appealing method to apply CSC-layer for robustness inference. It is great if this paper has a more detailed comparison of its time and memory for both training and inference with the other methods.

**Strengths And Weaknesses:**

Strengths:
This paper proposed an interesting and effective idea for robust inference with the proposed convolutional sparse coding layer and achieved impressive performance on both image corruption and adversarial attack.

The weaknesses mainly lay in the cost of the training and robustness inference:
1. This paper claims the fast training speed as one of its contributions. However, this needs careful comparison with its baselines.
2. Although this method does not require modifying the training procedure like the existing methods to obtain robustness, it has additional costs to get the optimal $\lambda$ for robustness inference.

---

> ### Author Response · Authors · 2022-08-02
> **Response to Reviewer nUHw**
>
> We thank the reviewer for the careful reading of our manuscript and the constructive remarks. Here we reply to those specific questions.
>
> >Q1: In Table 1, the model size of SDNet-18 and SDNNet-34 on CIFAR-100 are much smaller than on CIFAR-10, which seems weird.
>
> A: Thank you for the comments. It is indeed a typo. The model size of SDNet-18 and SDNet-34 on CIFAR-100 are 11.3M and 21.2M, respectively. We have fixed it in the revised version.
>
> >Q2: With similar performance, the proposed method is much faster than its baselines. In Table 1, the proposed SDNet only replaces the first convolutional layer with CSC-layer while SCN is a multilayer sparse coding network. For a fair comparison, this paper may compare the time and memory consumption of a single sparse coding layer between those methods.
>
> A: Following the reviewer’s suggestion, we replace the first convolution layer of ResNet18 with the sparse code layer from SCN[1], and keep the parameters the same as ResNet18 such as channel, strides, kernel size, etc. The comparisons of model size, test accuracy, memory used during training, and training speed are shown as follows:
>
> | CIFAR10      |    Model Size    |      Top-1 Acc   |   Memory      |  Speed |
> |---------------|-------------------|------------------|----------------|---------|
> |ResNet18     |   11.2M              |    95.54%       |   1.0GB         |1600 n/s |
> |SCN             |    0.7M               |   94.36%      |   10.0GB         |   39  n/s  |
> |SCN-first     |    11.2M             |     95.12%     |      3.5GB        |   158  n/s|
> |SDNet18      |    11.2M             |     95.20%     |      1.2GB        | 1500 n/s |
>
>
>
> | CIFAR100     |    Model Size    |      Top-1 Acc   |   Memory      |  Speed |
> |---------------|-------------------|------------------|----------------|---------|
> |ResNet18     |   11.2M              |    77.82%       |   1.0GB         |1600 n/s |
> |SCN             |    0.7M               |   80.07%      |   10.0GB         |   39  n/s  |
> |SCN-first     |    11.2M             |     78.59%     |      3.5GB        |   158  n/s|
> |SDNet18      |    11.2M             |     78.31%     |      1.2GB        | 1500 n/s |
>
> It can be seen that SCN-first is still much slower than our SDNet.
>
> >Q3: Each layer of CSC-layer of SDNet-18 and SDNet34 needs unrolling two iterations of FISTA and more iterations will only slightly improve the performance. As SDNet-18 and SDNet-34 have only one CSC-layer for the input images, I’m curious whether it is this low dimension (3 channels) of input that make two iterations sufficient. On SDNet-18-All and SDNet-34-All, could you list the dimension of the input and output of each CSC-layers and their corresponding iterations used?
>
> A: In SDNet18/34-All, the dimensions of the input and output of each CSC-layers are precisely the same as the one corresponded in ResNet 18/34, which are 3 -> 64 -> 128 -> 256 -> 512. And 2 FISTA iterations are used in all CSC-layers. We have conducted the ablation study on ImageNet, and we find that SDNet-18 with 2, 4, and 8 iterations of FISTA obtains 69.47%, 69.51%, and 69.79% Top-1 accuracy, respectively. While using more iterations slightly increases the model performance, it comes at the cost of increasing the training time and memory requirement as a result of unrolling of the FISTA algorithm. Hence, in all our experiments we use SDNet with 2 iterations. We will leave the exploration of adaptive FISTA iteration for different layers in the future work.
>
> >Q4: What is the cost to find the optimal lambda and calculate the residual in Algorithm 1?
>
> A: 1. For finding the relationship of optimal $\lambda$ as a function of $r$ (i.e., the function in Alg. 1, line 12), we need to perform #types_of_noise $\times$ #noise_levels forward processes for the training data (or a subset of the training data). Note that this only needs to be performed once and can be subsequently used for any test data.
> 2. For predicting the labels on a test set, we need to perform 2 forward processes for each data in the test set, where the first is used to obtain the residual and optimal $\lambda$, and the second is for obtaining final robust accuracy under the optimal $\lambda$.
>
> >Q5: I would like to confirm whether it is z of a CSC-layer that will be the input of the next layers.
>
> A: Yes. The sparse code of the CSC-layer will be the input of the next layer.
>
> >Q6: Does FISTA algorithm always randomly initialize z for any CSC-layer in any iteration during the training? If it is, is it possible to initialize it with the previous leaned values for an image when the model sees it again? This may reduce the number of FISTA iterations.
>
> A: In our work, the FISTA always initializes z from zeros for any CSC-layer during the training. Since data augmentation is used during training, the training data changes in different epochs even with the same input image. Hence, initializing z from those in previous epochs will not offer any benefits.

---

> > ### Comment · Reviewer_nUHw · 2022-08-09
> > **Thank you for the clarification**
> >
> > I thank the authors for the detailed response and experiments! I have increased my score. It would be great if the authors could further give out the computing time of Algorithm 1.

---

> > > ### Author Response · Authors · 2022-08-09
> > > **Thanks for your recognition of our work!**
> > >
> > > We thank the reviewer for acknowledging our work. To answer your question, we test the time cost on the CIFAR10-C dataset with 10000 samples using 1 GPU. As declared in our previous reply, we need to perform #types_of_noise $\times$ #noise_levels forward processes for the training data (or a subset of the training data) to find the relationship of optimal $\lambda$ as a function of $r$ (i.e., the function in Alg. 1, line 12). It takes about 50-100 forward processes (depending on the datasets), each of which costs about 5s. So the total time cost would be a few minutes. Once this stage finishes, the following prediction of the labels only takes 2 more forward processes, which is negligible.

---

> > > > ### Comment · Reviewer_nUHw · 2022-08-09
> > > > **Thanks for the clarification**
> > > >
> > > > Thanks to the authors for the clarification. I have no future questions. This is a great exploration to improve the robustness of the neural network to input perturbations by sparse coding. I agree with the reviewer EVjq that visualizing the learned sparse dictionary convolutional kernels will make it more insightful. Please include the discussions and the visualization in the final version.

---

> ### Author Response · Authors · 2022-08-07
> **Further discussion with the Reviewer**
>
> Dear reviewer nUHw:
>
> We thank you for the precious review time and valuable comments. We have provided corresponding responses and results, which we believe have covered your concerns. We hope to further discuss with you whether or not your concerns have been addressed. Please let us know if you still have any unclear parts of our work.

---

### Meta-Review · Area_Chair_j6dw · 2022-08-25

**Recommendation:** Accept
**Confidence:** Less certain

**Metareview:**

In this paper, the authors introduce a convolutional sparse coding layer, which is intended as a replacement for a convolutional layer that is has greater interpretability and stability.  Experiments show that a ResNet modified with this CSC-layer can achieve comparable performance on standard datasets as convolutional networks and are more robust to noise.  The strength of this paper is that the novel layer it proposes is faster than previous space coding networks and that it has comparable accuracy and speed to ResNets while being more robust.  A weakness of the paper is that the claims of improved interpretability do not transfer to the network as a whole.  The strengths of the paper outweighs the weaknesses, and the authors should clarify in the camera ready that interpretability is only intended layer-wise.

**Award:**

No

---

### Decision · Program_Chairs · 2022-09-14

Accept